# Developing a Machine Learning ‘Smart’ Polymerase Chain Reaction Thermocycler Part 2: Putting the Theoretical Framework into Practice

**DOI:** 10.3390/genes15091199

**Published:** 2024-09-12

**Authors:** Caitlin McDonald, Duncan Taylor, Russell S. A. Brinkworth, Adrian Linacre

**Affiliations:** 1College of Science and Engineering, Flinders University, GPO Box 2100, Adelaide, SA 5001, Australia; caitlin.mcdonald@flinders.edu.au (C.M.); russell.brinkworth@flinders.edu.au (R.S.A.B.); adrian.linacre@flinders.edu.au (A.L.); 2Forensic Science SA, GPO Box 2790, Adelaide, SA 5001, Australia

**Keywords:** PCR thermocycler, cycling conditions, machine learning, STR DNA profile

## Abstract

The introduction of PCR into forensic science and the rapid increases in the sensitivity, specificity and discrimination power of DNA profiling that followed have been fundamental in shaping the field of forensic biology. Despite these developments, the challenges associated with the DNA profiling of trace, inhibited and degraded samples remain. Thus, any improvement to the performance of sub-optimal samples in DNA profiling would be of great value to the forensic community. The potential exists to optimise the PCR performance of samples by altering the cycling conditions used. If the effects of changing cycling conditions upon the quality of a DNA profile can be well understood, then the PCR process can be manipulated to achieve a specific goal. This work is a proof-of-concept study for the development of a smart PCR system, the theoretical foundations of which are outlined in part 1 of this publication. The first steps needed to demonstrate the performance of our smart PCR goal involved the manual alteration of cycling conditions and assessment of the DNA profiles produced. In this study, the timing and temperature of the denaturation and annealing stages of the PCR were manually altered to achieve the goal of reducing PCR runtime while maintaining an acceptable quality and quantity of DNA product. A real-time feedback system was also trialled using an STR PCR and qPCR reaction mix, and the DNA profiles generated were compared to profiles produced using the standard STR PCR kits. The aim of this work was to leverage machine learning to enable real-time adjustments during a PCR, allowing optimisation of cycling conditions towards predefined user goals. A set of parameters was found that yielded similar results to the standard endpoint PCR methodology but was completed 30 min faster. The development of an intelligent system would have significant implications for the various biological disciplines that are reliant on PCR technology.

## 1. Introduction

Since its conceptualisation, the polymerase chain reaction (PCR) has played a fundamental role in our understanding of the world of deoxyribonucleic acid (DNA) around us. Despite the PCR having a wide range of applications across the fields of biology and medicine, the basic formula it uses for amplification has undergone little change since it was first introduced to forensic science in the early 1990s [1,2].

The development of a PCR method that enabled simultaneous amplification of four hypervariable regions, known as short tandem repeats (STRs), in the human genome marked the beginning of a transformative era in forensic science, an era where the value of DNA evidence skyrocketed [1,2]. Since then, the number of regions targeted for amplification has expanded from 6 in 1995 to 25 in 2018 [1,2,3], and there has been a significant decrease in the amount of DNA required to produce informative DNA profiles, which has been demonstrated on a single cell (0.006 ng [4], down from 2 ng in 1994) [5,6]. These developments have allowed DNA profiling to become a highly sensitive technique that affords high powers of discrimination between individuals. However, despite these advancements, the PCR cycling conditions used today are largely the same as those first used in the 1990s, with uniform conditions used in most forensic applications [2]. This uniform approach does not adjust or account for the variations in enzyme activity and amounts of target DNA across the PCR cycles, meaning there is potential for further optimisation in the system.

Trace DNA is commonly defined as low levels of DNA from which the biological source is unknown. DNA deposited from a touch is usually considered trace DNA. While trace DNA is one of the most common sample types encountered by operational forensic laboratories [7,8,9], it remains a sample type that has little success in producing informative DNA profiles [10,11]. Previous efforts have concentrated on pre-amplification of the whole genome [12], direct PCR of the sample [13], single-cell amplification [4] or reduced reaction volumes [14]. Despite the many advancements in the PCR for forensic science outside of cycling conditions, the limits of the technology are continually being pushed in an attempt to process smaller quantities, in more compromised conditions, and in shorter timeframes. One exception to the general lack of experimentation in PCR cycling conditions is in the area of speeding up reaction time within specialised equipment. Altered PCR protocols have been successfully integrated into forensic casework (such as in Rapid DNA); however, when these protocols are used with sub-optimal samples, the DNA profiles often contain undesirable characteristics, for example, small peak heights, heterozygote imbalance and dropout. Thus, there is still a need to overcome the challenges that trace samples present for DNA profiling [2,15].

Part 1 of this publication proposed a framework to produce a smart PCR system that uses machine learning to identify PCR program adjustments that can improve DNA profiling on a per-sample basis using real-time fluorescence feedback produced as the PCR proceeds. Theoretically, this system could be used to generate optimal DNA profiles from a range of challenging sample types (such as trace or degraded types and/or samples containing inhibitors) by amplifying each using individually tailored cycling conditions. A range of goals could be set for the system, such as to improve the quality of a profile (as just mentioned), or to reduce the cost (by altering PCR components or reducing volume while changing cycling conditions to maintain quality), or to improve efficiency (such as reducing the time taken to complete a PCR, again without a practically relevant reduction in profile quality). In this paper, we adjust PCR cycling conditions and apply the profile scoring metric for DNA profile comparison as outlined in the smart PCR framework in part 1 as a proof-of-concept implementation of the proposed framework. We initially chose to target PCR time reduction for the proof-of-concept implementation, for the following reasons:The variable that is tied directly to the goal (time) is an easily measurable quantity for which there can be no dispute (unlike a goal such as improved quality, in which there is subjectivity over the relative importance of different features, such as peak height vs. balance).There is a current interest in producing quick PCR programs, evident in the rise in the number and popularity of Rapid DNA instruments.There are existing alternate protocols being used in Rapid DNA instruments that can provide some guidance on which components of PCR programs can be altered and in what way.

In later work, we intend to explore other goals, such as improving DNA profile quality from low-level or degraded profiles or overcoming inhibition.

## 2. Method

In part 1, we provided an outline of all the components theoretically required to build a smart PCR system. In this study, we carried out a pilot study to demonstrate the performance of the theory to achieve the specific goal of reducing PCR runtime through a large-scale PCR cycling condition alteration experiment. Although we did trial the machine learning optimisation method proposed, the main aspect we did not test was the full training of an ANN-based statistical model (digital twin) of the PCR process. This was due to the absence of a large database upon which to train such a system. We intend to revisit this aspect later. As well as trialling some of the foundational components of the larger project in a manual way, the purpose of this study was to begin to build up a bank of real-time fluorescence data that could be used to train machine learning algorithms for PCR optimisation.

### 2.1. Ethics Approval

Ethics approval for this project was obtained from the Low-Risk Human Research panel of the Social and Behavioural Research Ethics Committee at Flinders University, Australia (reference number 4915). All DNA samples provided by a volunteer were obtained with informed consent.

### 2.2. Setting the Goal

We initially set the goal of reducing the time a PCR program takes to run with minimal loss of DNA profile quality. We chose this goal as it is relatively easy to achieve. By this, we mean PCR programs can certainly be changed in ways that guarantee they take less time. There is also literature relating to reduced PCR program times from use in Rapid DNA instruments [16,17] and other quick PCR program research [18] that could be consulted. A goal to reduce PCR program time also meant that the experiments could be conducted on optimal amounts of pristine DNA, which remained constant across all experiments.

### 2.3. Setting the Real-Time Feedback Mechanism

We trialled the use of a combined STR PCR and qPCR reaction mix as described by McDonald et al. [19]. The control DNA obtained and used in part 1 was also used as control DNA for all experiments in this study. The standard GlobalFiler (ThermoFisher Scientific, Scoresby, VIC, Australia) PCR components comprised 7.5 µL of master mix, 2.5 µL of primer mix and 15 µL of DNA in solution (made up to a concentration so that 500 pg of DNA was used in the PCR). The combined kit used 7.5 µL of GlobalFiler master mix, 2.5 µL of GlobalFiler primer mix, 4.5 µL of Investigator Quantiplex Pro master mix, 4.5 µL of Investigator Quantiplex Pro reaction mix and 6 µL of DNA in solution (made up to a concentration so that 500 pg of DNA was used in the PCR).

We compared the results obtained from the combined quantitation/STR PCR system described above with those of a two-tube system where Quantiplex Pro reaction mixture was set as per the manufacturer’s instruction (but at half volume so as to align with the half volume in the combined kit and allow direct comparison of performance), and a GlobalFiler reaction was set up as per the manufacturer’s instructions. These two reactions were amplified separately but at the same time, both using the manufacturer’s recommended 30-cycle GlobalFiler PCR protocol. The PCR products were separated on a 3500 Genetic Analyzer^TM^ (ThermoFisher Scientific) using 8.5 μL Hi-Di Formamide, 0.5 μL 600 LIZ^®^ Size Standard (ThermoFisher Scientific) and 1 μL of amplified PCR product. The 3500 settings were 1.2 kV/15 s injection, 13 kV/1550 s runtime, dye set J6.

### 2.4. Calculating the Profile Score Metric

In part 1, we defined a score, *f*(*P*), for a DNA profile *P* with allele peak heights $x_{1}$, …, $x_{n}$ by
(1)$$f\left(P\right)=K_{0}[\log_{10}\left[p\left(\bar{x}|\mu_{a},\sigma_{a}\right)\right]-C+K_{1}\log_{10}\left[p\left(c_{v}|\lambda \right)\right]+K_{2}n_{a}+K_{3}t$$
where the following definitions apply:

$\bar{x}=\frac{1}{N}\sum_{i=1}^{n}x_{i}$ is the mean of all observed allelic peaks;

$p\left(\bar{x}|\mu_{a},\sigma_{a}\right)∼N\left(\mu_{a},\sigma_{a}\right)$ is the probability density function at the point $\bar{x}$ for the distribution with mean $\mu_{a}$ (the ideal or reference mean value) and standard deviation $\sigma_{a}$ (the value for weighting variations from the ideal);

${C=\log}_{10}\left[p\left(\mu_{a}|\mu_{a},\sigma_{a}\right)\right]+{log}_{10}\left(\lambda \right)$ is the constant offset to ensure an ideal peak height achieved a score of 0;

$p\left(c_{v}|\lambda \right)∼\exp\left(\lambda \right)$ is the exponential distribution at the point $c_{v}$ with coefficient λ;

$c_{v}=\frac{\sqrt{\frac{1}{N}\;\sum_{i=1}^{n}{\left(x_{i}-\bar{x}\right)}^{2}}}{\bar{X}}$ is the coefficient of variation in the peaks;

$n_{a}$ is the number of observed artefact peaks;

t is the time (in minutes) the PCR program takes to complete;

*K* parameters weight the importance of each element of the score.

All calculations and production of graphs were carried out in R [20].

With reference to Equation (1), we used values $K_{0}=1$, $\mu_{a}=5000$, $\sigma_{a}=1000$, λ=1 and $K_{2}=-2$ The choice of parameter values for the target peak height ($\mu_{a}$) and standard deviation ($\sigma_{a}$) was subjective and based on the personal experience of the authors. Based on the selection of $\mu_{a}$ and $\sigma_{a}$, the peak height offset value (that resulted in a baseline value of 0) C was calculated to be −3.399. The value for the COV (λ) was largely irrelevant as it scaled linearly by the gain value ($K_{1}$). During the pilot parameter tests, the average score for the peak height component was found to be −2.047. Using this as a target, $K_{1}$ was set to 5 to ensure, on average, an equal contribution from the COV component of the score based on the pilot study (−2.083). The value chosen to penalise any artefacts present in the profiles ($K_{2})$ was subjectively set to −2 to ensure that profiles containing artefacts were harshly penalised. Three different values were used for the time component, $K_{3}=\left\{0,-0.01,-0.05\right\}$, representing no, small and large weightings. The value of −0.05 gave, on average, from the pilot study, a score of −2.191, which was approximately equal to the height and COV components of the score. This meant that, when time had a high weighting, the three main quality components (allele peak height, allele peak height variation and running time) were approximately equally weighted, while a single artefact in the data would contribute approximately equally to any one of the other components.

### 2.5. Trialling Different PCR Program Alterations

PCR program alterations were either carried out on the denaturation step or the annealing/extension step of the standard GlobalFiler PCR program. In addition, changes were either made in a way that altered each cycle in the same way (referred to as ‘bulk’ changes) or made in a way that was incremental across the program (referred to as ‘gradient’ changes). Table 1 shows the different set-ups of PCR programs in parameterised form. Note that the one-step changes (altering only one aspect of the PCR program) were carried out first, and then the final two-, three- or four-step changes (altering multiple aspects of the PCR program together) were carried out after review of the initial one-step change results.

Note that denaturation temperature program changes were trialled even though they did not change the timing of the PCR program. This was carried out as the interaction between temperature and timing changes was unknown, and it was hypothesised that a poor-quality profile produced using a quicker PCR program may be improved by a temperature change.

A previous study explored the viability of using published Rapid DNA programs for non-rapid PCR kits and found that the STR profiles generated were of very poor quality [21]. It is hypothesised that the components within Rapid DNA PCR kits are optimised for these rapid programs; thus, when non-rapid kits are used with Rapid DNA PCR programs, they produce poor DNA profiles. As a result, we did not use cycling conditions from published Rapid DNA PCR programs for this study; instead, we made small, controlled changes in the PCR cycling conditions from a standard program. This allowed us to acquire knowledge of how each of these small changes influenced DNA profile quality, which can be used as training data in subsequent projects.

### 2.6. Weighing Time against Profile Quality

Profiles’ scores were considered in three ways: unadjusted, considering time as mildly important and considering time as highly important. The unadjusted values were the raw profile quality scores obtained by inspection of the DNA profiles (*K*_3_ = 0 in Equation (1)). When considering time as being mildly important, the raw profile quality score was adjusted by subtracting 0.01 for each minute the program ran (*K*_3_ = −0.01 in Equation (1)). When considering time as being highly important, the raw profile quality score was adjusted by subtracting 0.05 for each minute the program ran (*K*_3_ = −0.05 in Equation (1)).

### 2.7. Comparing Profile Quality between PCR Programs

The quality of DNA profiles generated using the various altered PCR programs was compared to the quality of the profiles generated using the standard GlobalFiler conditions. The quality of DNA profiles produced using the combination kit was compared to those produced using GlobalFiler only for each of the altered PCR programs and the standard GlobalFiler program. All comparisons were conducted using *t*-testing in R [20], where *p*-values less than 0.05 were considered significant.

### 2.8. Using Experimental Results to Suggest New PCR Cycling Conditions

At the beginning of the experiment, there was limited information about how changes in PCR cycling conditions may affect profile quality. Therefore, the PCR cycling condition changes were made by parameterising the conditions of the cycling program, and these were altered in a controlled manner to build a bank of data which could be used to inform further choices. Figure 1 shows typical PCR program cycling conditions, parameterised so that each time, *t*, and each temperature, *C*, for each step of a cycle (i.e., denaturation, annealing and extension) have their own terms. At the most configurable, this leads to 6*N* parameters for an *N* cycle program. Such a large number of parameters is likely to take too many experiments to optimise individually and so parameter values can be tied to each other to reduce the dimensionality of the problem. At the most extreme, for the task of reducing PCR program time, the problem could be reduced to a single dimension by scaling the time of each step down proportionally.

In Figure 1, for step *X* in cycle *N*, the temperature is *C_X,N_* and is held for *t_X,N_* seconds. We parameterised the times and temperatures using the following linear equations:(2)$$\mathrm{Denaturation}\;\mathrm{time}:\;{t_{D,n}=\beta }_{t,D,0}+\beta_{D,1}N$$
(3)$$\mathrm{Denaturation}\;\mathrm{temperature}:\;{C_{D,n}=\beta }_{C,D,0}+\beta_{C,D,1}N$$
(4)$$\mathrm{Annealing}/\mathrm{extension}\;\mathrm{time}:\;{t_{AE,n}=\beta }_{t,AE,0}+\beta_{t,AE,1}N$$
(5)$$\mathrm{Annealing}/\mathrm{extension}\;\mathrm{temperature}:\;{C_{AE,n}=\beta }_{C,AE,0}+\beta_{C,AE,1}N$$

Note that we combined the annealing and extension into a single term compared to Figure 1 given that the GlobalFiler PCR cycling conditions only have two steps, a denaturation and an annealing/extension step. Using the linear equations, the time and temperature could be changed so that the same time (or temperature) is used in each step (referred to a as bulk change) by setting the slope terms, $\beta_{.,.,1}=0$. Alternatively, gradient changes can be achieved by setting a non-zero value for the slope term. For example, if the denaturation step time is being considered, then Equation (2) is used. If the starting time is 10 s, then $\beta_{t,D,0}=10$. If no change to this timing is desired across the PCR program, then the value for $\beta_{D,1}$ would be set to 0, and then, for all cycles, $t_{D,n}=10$. If a gradient is desired, such as a 1 s decrease every 5 cycles, then $\beta_{D,1}=-0.2$. This would lead to the GC1 denaturing timing conditions seen in Table 2.

For each of the programs listed in Table 1, we had five replicate profiles. We ignored the annealing/extension temperature parameters as they were not altered in any of the PCR program changes trailled. Therefore, we had 85 profiles to preliminarily explore the six-variable parameter space.

Once initial information about the system performance over the parameters’ space (as detailed in Table 2) was collected, it was necessary to use that information to perform optimisation to discover the optimal set of parameters for a given situation. The method chosen to perform this was based on the firefly algorithm [22]. Conceptually, each set of parameters could be thought of as a firefly, flying around in six-dimensional space. Each set of parameters leads to a DNA profile produced in a specific time, and, ultimately, a score, which, in the firefly analogy, is related to the fly’s brightness. Flies are attracted to brighter areas (high scores) and repelled from darker areas (low scores), and so, for each iteration, each fly will move according to the brightness of the fireflies around it. The more distant the brightness, the weaker the effect. For each set of parameters (e.g., a single firefly) going into the algorithm, a set of parameters will be obtained after an iteration of movement (firefly at a new location). See Figure 2.

Specifically for our application, the following process was undertaken:(1)Sensible bounds were placed on each parameter so that the algorithm did not suggest values that would clearly not work. This step was only required due to the low number of datapoints, and, in larger datasets, the bounds on parameter values would be self-imposed due to lower scores obtained as the parameter values become more extreme. The bounds placed on the parameters were $1\;\leq \beta_{t,D,0}\;\leq 15$, $-0.4\leq \beta_{t,D,1}\leq 0.4$, $87\leq \beta_{C,D,0}\leq 95$, $-0.3\leq \beta_{C,D,1}\leq 0.3$, $25\leq \beta_{t,AE,0}\leq 100$, $-1.5\leq \beta_{t,AE,1}\leq 1.5$. These limits were put in place to prevent parameters such as negative times or excessive temperatures that could damage the DNA.(2)The supplied parameters were used in the PCR process. Performance scores were generated from the results based on Equation (1).(3)Each parameter value was scaled (normalised) to be between 0 and 1 so that each parameter had the same level of optimisation within the algorithm.(4)The performance score for each parameter set was normalised between −1 and 1 to ensure high values were attractive and low values repulsive.(5)A learning rate (which dictates the size of the move that can be made in each iteration) of 0.5 was set. If a very large number of repetitions were undertaken, this level could be reduced after a number of rounds to better hone in on the optimal parameters.(6)For each point of focus, the parametric distance to each other point was calculated.(7)The movement of the point of focus from each other point was set as the difference between the current parameter position of the point of focus and other parameter sets, multiplied by the learning rate, multiplied by the score, and scaled by one less than the number of datapoints.(8)The new positions were then rescaled back to their natural form and represented new sets of parameter values.(9)The new parameter values were tested, performance scores determined, and the cycle of the process could be started again with the new parameters.

When many points exist, they can be ranked based on the theoretical time the program would take to run; however, we chose to run one iteration of the firefly algorithm to produce a full set of new values that could be trialled.

## 3. Results

### 3.1. Combined GlobalFiler and Investigator Quantiplex Pro Results

DNA profiles were generated from the standard GlobalFiler-only PCR and the combined PCR mixture (the qPCR and PCR combined kit) when separated by capillary electrophoresis. In general, there was a loss of peak height for the combined kit. Figure 3 shows the distribution of height differences between the combined kit and standard kit. The distribution is modelled on the per-profile average peak height of the DNA profiles produced by the GlobalFiler-only kit divided by the average peak height of the DNA profiles produced by the combined kit. While the combined kit was found to produce DNA profiles with lower average peak heights than the profiles produced using GlobalFiler only (approximately 1.6 times or log_10_[0.21]), this variation was found to be insignificant (*p* = 0.077, shaded area in Figure 3).

The Standard Quantiplex Pro reaction (Figure 4) was run for 40 cycles. The red dashed point in Figure 4 shows the 30-cycle mark, which corresponds to the endpoint of a standard GlobalFiler PCR program. It can be seen that the inflection point of the quantification reaction has not been reached. There is a risk that even using a two-tube system (where the quantification and STR amplification reactions are in separate vessels) without 10 cycles of pre-amplification of the quantification reaction prior to the start of the accompanying GlobalFiler PCR will not provide useable real-time information. This fact is shown in Figure 4, where the quantification reaction is depicted under standard qPCR conditions, and the curve only just starts to visibly increase in the final cycles (>23). It is yet to be seen whether there is sufficient information in the real-time fluorescence in this combined system to facilitate learning and, eventually, real-time program adaption. However, what is clear is that the PCR system developed using a combined GlobalFiler/Quantiplex Pro reaction did provide some real-time fluorescence feedback that could be exploited.

Furthermore, the fluorescence data obtained from the GlobalFiler/Quantiplex Pro combination were compared to the fluorescence data obtained using the two-tube method, where the GlobalFiler and Quantiplex Pro reactions were conducted separately. The resulting amplification curves (Figure 5) indicate that the STR kit does produce some detectable fluorescence when used independently of the qPCR reagents. However, as the primers in the STR kit will fluoresce regardless of if they are incorporated or unincorporated, the amount of fluorescence produced by them should remain constant across the PCR. This means there is the potential to correct for any fluorescence produced by the STR primers in the first few cycles to significantly reduce if not eliminate the interference of the STR kit in the real-time fluorescence feedback we aim to acquire.

In Figure 5, it can be seen that the Quantiplex Pro system alone produced very little fluorescence (which was smoothed to a flat line by the model). Despite this, the fluorescence from the combined Quantiplex Pro and GlobalFiler system was markedly more intense than the GlobalFiler alone. It is not currently clear whether there was some interaction occurring to cause this unexpected jump in fluorescence in the combined system. The performance of the real-time fluorescent feedback under varying conditions is the next phase of research to be conducted in our work.

### 3.2. Results of Altering PCR Program Conditions

#### 3.2.1. One-Step Changes to Denaturation Timing

When profile quality was considered in isolation, the PCR protocols involving bulk changes to denaturation timing using the combination kit produced DNA profiles that were significantly lower in quality than profiles produced following the standard GlobalFiler protocol (Figure 6). The PCR protocol involving a gradually decreasing denaturation timing (gradient change 1 protocol) with the combination kit produced profiles of significantly lower quality than the standard protocol, but generally higher quality than the other altered protocols.

Consideration of the quality of the DNA profile, and incorporating the timing penalty, led to a ‘score’ (based on Equation (1)). When the PCR runtime was considered mildly important in quality assessment, the programs with altered denaturation times using the combination kit were up to 29 min faster. The bulk change programs, and one gradient program (gradient change 2 protocol), still produced profiles with a significantly lower score than the standard. Notably, the gradient change 1 program produced profiles that were approximately equal in quality to those produced using the standard protocol (Figure 6). Considering the PCR runtime as highly important when scoring found that profiles produced using both bulk change programs and the gradient change 2 program still yielded significantly lower scores than the standard profiles. However, the profiles produced using the gradient change 1 program were now of significantly higher quality than the profiles produced following the standard protocol.

DNA profiles produced using a denaturation time of 5 s (bulk change 1 protocol) with only the GlobalFiler kit were not of significantly different quality to those produced using the combination kit for the same protocol across all scores (see Appendix A). Similarly, the profiles produced using a gradually decreasing denaturation timing (gradient change 1 protocol) or gradually increasing denaturation time (gradient change 2 protocol) with GlobalFiler only were of approximately equal quality to those produced using the combination kit under the same conditions. However, the profiles generated using a denaturation time of 2 s (bulk change 2 protocol) with the GlobalFiler kit were notably lower in quality than the profiles produced using the same PCR program with the combination kit across all scores (see Appendix A).

However, when the scores were standardised, the DNA profiles produced using the GlobalFiler kit were not significantly different in quality to those produced using the combination kit for the same protocol across all standardised scores (see Appendix A).

#### 3.2.2. One-Step Changes to Annealing/Extension Timing

The DNA profiles generated using an annealing/extension time of 60 s (bulk change 1 protocol) and 30 s (bulk change 2 protocol) with the combination kit were significantly lower in comparison to the profiles produced under the standard GlobalFiler conditions (*p* = 0.0216 and *p* = 0.0122, respectively) (Figure 7). The profiles generated with a PCR program involving a gradual increase or decrease in annealing/extension timing (gradient change protocols 1 and 2) using the combination kit were not found to have a significant variation in raw quality score when compared to the standard profiles (see Appendix A).

When considering the PCR runtime as mildly important, the influence of the bulk and gradient changes to annealing/extension timing on overall profile score was largely conserved despite the altered protocols being up to 50 min faster than the standard protocol. The DNA profiles generated using the two gradient change programs were still approximately equal in score to those generated using the standard program. While the profiles produced using the bulk change 2 protocol were still of significantly lower score than the standard profiles (*p* = 0.0122), the profiles produced using the bulk change 1 protocol were found to be of approximately equal score to the standard (Figure 7).

When PCR runtime was considered to be highly important, the bulk change 1 protocol produced DNA profiles that were of significantly higher score than the profiles generated using the standard GlobalFiler protocol (Figure 7). The profiles generated using the gradient change protocols were still found to be of approximately equal score to the standards, and the profiles generated using the bulk change 2 protocol had a significantly lower score than the standards (*p* = 0.0122).

The profiles generated using a gradually decreasing annealing/extension time (gradient change 1 protocol) with the combination kit were found to be of approximately equal quality to those produced using GlobalFiler only for the same protocols (see Appendix A). However, the DNA profiles produced using an annealing/extension time of 60 s (bulk change 1 protocol), an annealing/extension time of 30 s (bulk change 2 protocol) and a gradually increasing annealing/extension time (gradient change 2 protocol) with only the GlobalFiler kit were found to be of significantly lower quality than those produced using the combination kit for the same protocol across all scores (*p* = 0.033, *p* = 0.00038 and *p* = 0.0316).

When the scores were standardised, the DNA profiles produced using the gradient change programs and the GlobalFiler kit were not significantly different in quality to those produced using the combination kit for the same protocol across all standardised scores (see Appendix A). However, despite standardisation, the profiles produced using a reduced annealing/extension time of 30 s with GlobalFiler only were still found to be of significantly lower quality than those produced using the combination kit for the same protocol across all scores (*p* = 0.0117).

#### 3.2.3. One-Step Changes to Denaturation Temperature

All DNA profiles generated using an altered denaturation temperature with the combination kit were found to be significantly lower in quality than the profiles produced under the standard GlobalFiler conditions (Figure 8). However, while the DNA profiles produced using a gradient increase in denaturation temperature (gradient change 2 protocol) were significantly lower in quality than the standard profiles, of all the altered protocols trialled, these were closest in quality to the standards. The DNA profiles generated using the combination kit were not significantly different in overall quality than the DNA profiles generated using only the GlobalFiler kit for all programs involving an altered denaturation temperature.

The DNA profiles generated using a gradually increasing denaturation temperature with the combination kit were found to be of approximately equal quality to the profiles produced under the standard GlobalFiler conditions (Figure 8). The DNA profiles generated using a denaturation temperature of 90 °C (bulk change protocol 1) and 88 °C (bulk change 2 protocol) with the combination kit were significantly lower in comparison to the profiles produced under the standard GlobalFiler conditions (*p* = 0.0122 for both) (Figure 8). The profiles generated using a gradually decreasing denaturation temperature with the combination kit were also found to be significantly lower in quality when compared to the standards (*p* = 0.0122) (Figure 8).

DNA profiles generated using the combination kit were not significantly different in overall quality to the DNA profiles generated using only the GlobalFiler kit for all programs involving an altered denaturation temperature (see Appendix A). When the quality scores were standardised, the profiles produced using the bulk change programs and gradient change 1 program with the combination kit were of significantly lower quality than the profiles produced using the same protocols with GlobalFiler only across all scores (Appendix A). However, the profiles generated using a gradually increasing denaturation temperature (gradient change 2 protocol) with the combination kit were of significantly higher quality than the profiles generated using GlobalFiler only for the same protocol across all standardised scores (Appendix A).

#### 3.2.4. Two-Step Changes

When total runtime was not considered, the profiles generated using the optimised protocols with the combination kit were lower quality than those produced following the standard GlobalFiler protocol (Figure 9). When the profile scores were adjusted to include the PCR runtime as mildly important, the variation in quality between profiles produced using the optimised protocols and the standard GlobalFiler protocol with the combination kit was maintained. However, out of all the optimised protocols trialled, the profiles generated using protocol version 3 were closest in quality to the profiles generated using the standard protocol (Figure 9). This can be attributed to optimised protocol version 3 being 30 min faster than the standard GlobalFiler protocol.

Lastly, when the scores were adjusted to make the runtime highly important, the profiles generated using optimised protocols 1, 2 and 3 were all of approximately equal quality to the profiles produced following the standard GlobalFiler protocol. This can be attributed to these optimised PCR programs having runtimes that were 28–31 min faster than the standard.

Importantly, the DNA profiles produced using the combination kit were also found to be of significantly higher quality than the profiles generated using only GlobalFiler for all of the optimised PCR programs trialled (see Appendix A). When standardised, the DNA profiles produced using optimised protocols 2, 3 and 4 with the combination kit were all still of significantly higher quality than the profiles produced using GlobalFiler only, but profiles produced using optimised protocol 1 with the combination kit and GlobalFiler were found to be of approximately equal quality (see Appendix A).

It is important to note that all optimised PCR programs trialled produced some DNA profiles with instances of dropout; however, the profiles still contained information sufficient for downstream interpretation and analysis. The potential exists to alter the profile metric to make the peak height component more dominant than in this trial. Given a database now exists of all the profiles generated, it is possible to re-evaluate the dataset and trial various combinations of peak height and peak variability model parameters to identify which combination is most intuitively correct.

### 3.3. Suggesting New PCR Cycling Conditions Using Machine Learning

Overall, this provided 85 profiles’ worth of real-time fluorescence data, trialling 17 different combinations of PCR cycling conditions (five repeats of each of the combinations), that could be used in the training of a machine learning algorithm. Before the real-time data were used, we demonstrated the use of machine learning to suggest new parameters based on the DNA profile scoring results obtained.

The firefly algorithm was used, incorporating all 17 combinations. Initial trials showed a sub-optimal performance of the firefly algorithm due to the fact that not enough sample space had been explored in the dataset of 17 points. To rectify this, additional parameter sets were developed by generating 60 random values for each of the six parameters ($\beta_{t,D,0}$, $\beta_{t,D,1}$, $\beta_{C,D,0}$, $\beta_{C,D,1},\;\beta_{t,AE,0},\beta_{t,AE,1})$ from truncated normal distributions. The constraints used for each parameter aligned with the bounds for that specific parameter, as identified in Section 2.7. From the 60 random sets of parameters generated, the combinations that pushed the cycling parameters outside of the acceptable ranges were removed. This led to the identification of six additional PCR programs using the randomly generated parameter values shown in Table 3.

These six randomised parameter sets were then trialled to generate DNA profiles that could be scored and fed back into the system. This increased the sample space explored and thus worked to improve the performance of the algorithm. The results of these randomised PCR programs can be seen in Figure 10. As expected, all randomised PCR programs gave significantly lower scores than the standard; however, the point of producing these programs was not to obtain high scores, but rather to fill out some of the parameter space to inform the firefly algorithm (in this case, with very dim flies). Hence, the system knew what parameters not to use.

Using the 17 original sets of parameter values and the 6 random sets of values, the firefly algorithm was used and suggested the values shown in Table 4.

To complete the proof of concept, there were four sets of conditions suggested by the firefly algorithm that were trialled (those rows bolded in Table 4 and marked with an asterisk). The four sets of parameters were chosen to be trialled based on their theoretical success, previous experimental results and the projected impact of the time differences on the scores. Suggested programs S9 and S17 were chosen as they were expected to produce good-quality DNA profiles, while suggested programs S2 and S7 were chosen as they were expected to produce lower-quality DNA profiles. The results of these suggested methods can be seen in Figure 11.

## 4. Discussion

The goal for the PCR program alterations was to reduce PCR time without significantly reducing profile quality. While the practical outcome of the experiment was not hugely successful (i.e., in producing a program that ran quicker than the standard and with higher-quality profiles), this is secondary, as the main goal of the study was to start practically applying the theoretical steps outlined in part 1 of the publication to train a smart PCR system. In addressing these theoretical points, we were able to achieve the following:Create a combined quantification/STR PCR reaction that still produced high-quality DNA profiles;Carry out complex PCR program manipulations using an open qPCR instrument;Utilise a profile-scoring equation to assess the outcome of the PCR, and weigh it against a goal (in this case, PCR program time);Take a set of experiments and build a statistical framework for predicting new PCR program parameter values to trial;Trial new predicted PCR parameters.

Given the results of our pilot study, we found that a combined kit system does produce profiles of reasonable quality and can provide limited real-time fluorescence feedback. Depending on the quality of the real-time fluorescence data provided by the combined kit, it might be necessary to use a two-tube system. In this set-up, a quantitative PCR and STR PCR would be performed simultaneously but in separate tubes. The quantitative PCR should be primed with a number of initial cycles to ensure it provides meaningful real-time data at the beginning of the STR amplification process. There are limitations to using a split-tube system, the main one being that there is a possibility that the fluorescence data obtained from the qPCR tube would not exactly reflect the amplification efficiency within the STR PCR tube. Thus, the capability to monitor the exact amplification kinetics that are afforded by use of the combination kit would be lost.

In our pilot study, we ‘unsmartly’ altered PCR program conditions (in other words, not reacting to real-time feedback). We found that, when using a combined kit system, the goal of shortening PCR program time, without compromising DNA profile quality, can be approached. Of the programs trialled, the best altered performance was obtained using a gradually decreasing denaturation time (94 °C for 10–5 s) and a gradually increasing annealing/extension time (59 °C for 90–115 s). Depending on the weight given to the timing component, these modifications produced DNA profiles with higher scores than those produced using the standard GlobalFiler PCR protocol, with a time saving of 30 min. Using these results, we were able to use a machine learning firefly algorithm to suggest new conditions for reaching the goal of a quicker PCR program. The success of this final step was limited due to the fact that none of the suggested parameter set values produced profiles with higher scores than the originally trialled sets. Nevertheless, it showed the conceptual ability to use machine learning to achieve the desired goal, and it is likely that the issue in our proof of concept was merely one of low sample size and limited rounds of suggesting new parameter values. This is particularly true given that machine learning/optimisation algorithms typically require many iterations (generations) before yielding good results [23,24,25].

This study focused on optimisation using standard forensic profiling kits and laboratory hardware. This was to ensure the optimised PCR programs could be directly applied to casework in operational forensic laboratories. However, the findings of this study could be trialled with a range of other commonly used forensic profiling kits (e.g., VeriFiler Plus), quantification kits (e.g., Quantifiler Trio), detectors, fluorophores, DNA polymerases and buffer compositions. When considered alongside the findings of other recent studies [19,21], there is the potential for additional optimisation to complement the PCR programs by altering the make-up of these forensic profiling kits. Whilst the combination kit was found to be successful, if it is not possible to deconvolute the fluorescence data obtained from it, the composition of the kit may need to be further refined, or an entirely new bespoke kit with dual quantification and STR amplification capabilities may need to be designed. There is also the possibility to make changes to a qPCR instrument in future studies, such as in the wavelength of the detectors being used.

Optimisation of STR data from trace samples is core to the concept of a smart PCR machine rather than the analyses of reference samples. Trace DNA, while the most common sample type submitted to a forensic science laboratory [7,8,9], generates poor genetic information [10,11]. Increasing the genetic data from trace DNA inevitably will provide greater DNA data in criminal case investigations. It is only the timing at stages within the amplification that is being modified and not an overall increase in the number of cycles, with the inherent introduction of stochastic effects [26], none of which was recorded in this study. An increase in alleles amplified yet with no stochastic effects increases the opportunity to transfer into operational use the concept described in this study. As a parallel example, the RapidHIT^®^ID system was developed for reference samples but more recently has been applied to trace samples typical of those encountered in forensic science [4,27]. A difference between the technologies is that the chemistry and cycling parameters in RapidHIT^®^ID remain constant whether the sample added to the cartridge is a buccal swab or a swab from an item containing trace DNA [27], whereas the smart PCR concept allows modifications to the cycling parameters.

In this experiment, we were able to complete what would be considered a single round of training a machine learning algorithm. Given the complexity of the system on which this system is being applied, it is likely that multiple rounds of training would be required before significant advances are observed. It may also be the case that the current standard conditions do in fact happen to be the best cycling conditions (at least without also adjusting the reaction chemistry, as is performed for the Rapid DNA PCR kits). With increased data from the performance of PCR cycling conditions, it may be that different, or more sophisticated, methods of machine learning could be used. One possibility is to model the performance of the PCR system with an algorithm such as an artificial neural network, building up a digital twin that can then be probed for new cycling conditions. Another possibility is that the optimal settings for DNA replication vary based on the current concentration and that a single evaluation of a quality score at the end of the process is not the best way to evaluate the system. Instead, a better system may look at the rate of change in fluorescent feedback obtained during DNA replication by using a qPCR machine. This real-time feedback will likely give more insight and lead to better, and more variable, parameter sets for use.

## 5. Conclusions

We outline in this work the basic considerations for a smart PCR system that could have wide-reaching impacts on genetic laboratory work. The ultimate goal would be to produce a PCR system that could respond on a cycle-by-cycle basis as the program progresses in order to amplify each sample to its optimum. This is a lofty goal, and there are many fundamental aspects to producing such a system that need to be developed along the way. In part 1 of the publication, we outlined the theory for addressing some of these fundamental aspects, and, in part 2, we have shown how those theoretical aspects can be put into practice. Like almost any machine learning task, the key is to have a large database of data to train a system. In the initial proof of concept, we chose a simplified version of the main goal, that is, to train an open-loop system (rather than a system that changes in response to real-time feedback), and to employ a relatively simple machine learning algorithm, the firefly algorithm. We were successful in showing how an open-loop smart PCR system could be developed and refined over time and plan to expand the sophistication of the machine learning approach as more data are collected from these initial experiments.

## Figures and Tables

**Figure 1 genes-15-01199-f001:**
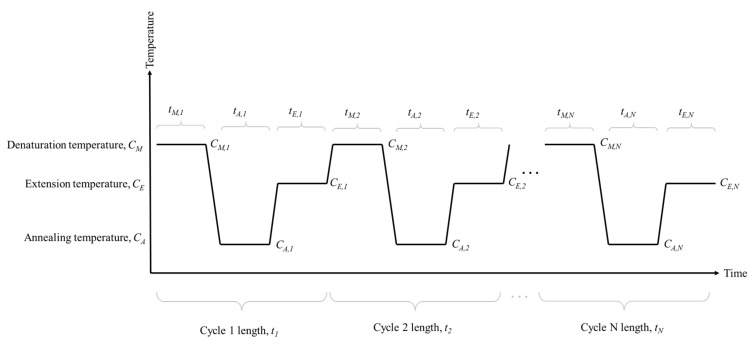
Parameterised depiction of a typical PCR program. For each cycle, the temperature is defined by a parameter, C, and the time each stage is held for is defined by parameter t. By parameterising the program in this way, the cycling conditions can be adjusted in a constant way across the whole program, or in a gradient manner, with the value dependant on the cycle.

**Figure 2 genes-15-01199-f002:**
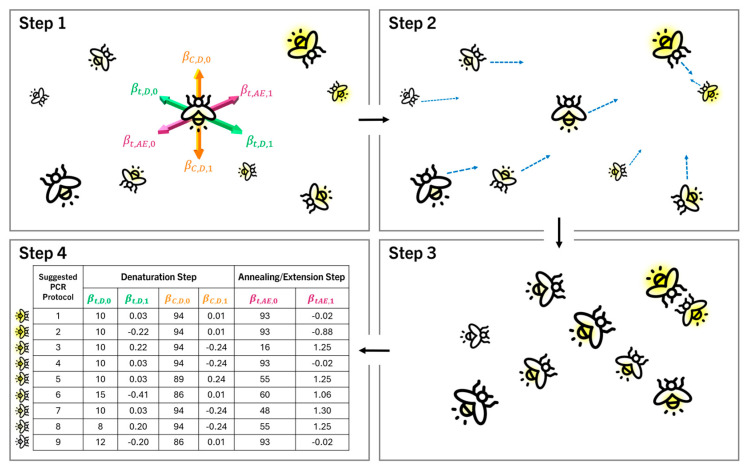
Diagrammatic explanation of the firefly algorithm used to suggest new PCR programs in this study. Step 1 (**top left**): All fireflies begin in the 6-dimensional (6-D) PCR parameter space according to their PCR parameters and glowing based on DNA profile quality. Step 2 (**top right**): After each round of testing, each firefly moves towards the PCR parameters that produce the best DNA profiles, and away from those that produce less desirable profiles. Step 3 (**bottom right**): All fireflies now have new, more attractive positions in the 6-D PCR parameter space that should produce better DNA profiles. Step 4 (**bottom left**): The new firefly positions are converted back to the optimised PCR cycling conditions they represent for further testing.

**Figure 3 genes-15-01199-f003:**
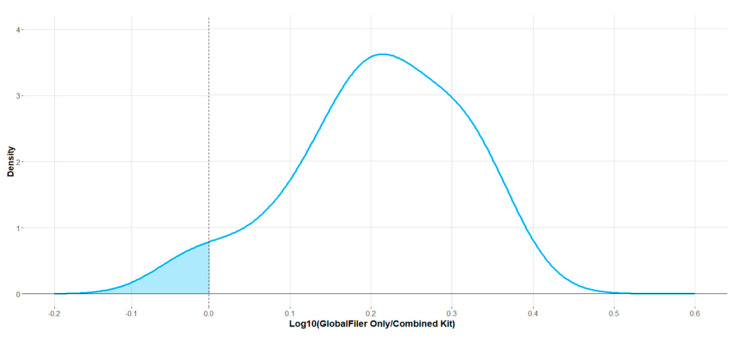
The experimental distribution of the average peak height of the DNA profile produced by the GlobalFiler-only kit divided by the average peak height of the DNA profile produced by the combination kit. A vertical dashed line at log_10_ = 0 indicates the point of no difference in average peak height between the combined kit and GlobalFiler-only kit. The shaded area shows the area of the distribution which is on the left side of 0. This relates to 7.7% of the area of the curve, which can be used as a measure of the significance of the difference being plotted.

**Figure 4 genes-15-01199-f004:**
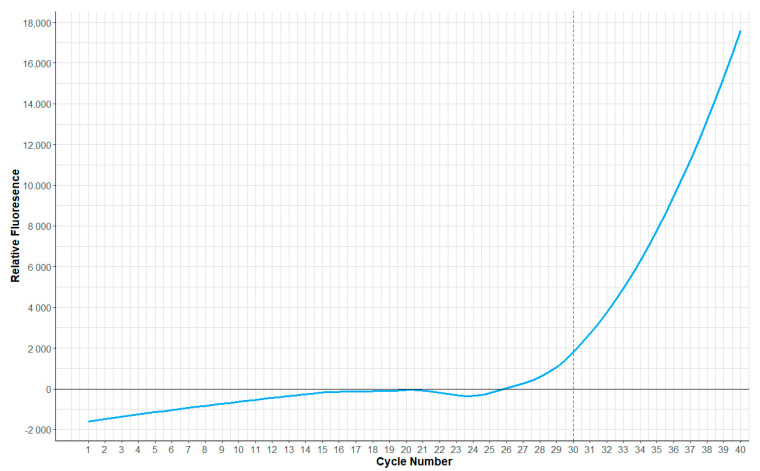
Real-time fluorescence feedback obtained from the Chai Open qPCR Instrument and 0.5 ng of starting DNA showing the amplification curve generated using the Investigator Quantiplex Pro Kit following the manufacturer’s 40-cycle protocol. The red dashed line highlights the 30-cycle mark at which the standard 30-cycle GlobalFiler protocol ends.

**Figure 5 genes-15-01199-f005:**
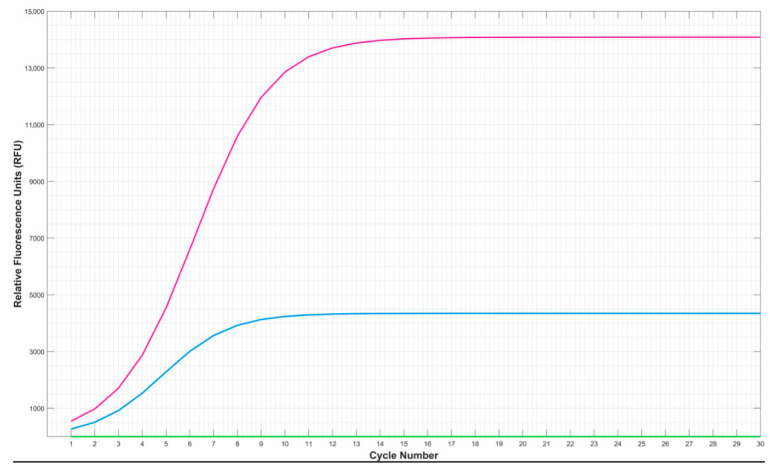
Real-time fluorescence feedback obtained from the Chai Open qPCR Instrument and 0.5 ng of starting DNA showing the amplification curves generated using the GlobalFiler and Investigator Quantiplex Pro^®^ RGQ Combination (pink), the Investigator Quantiplex Pro Kit on its own (green) and the GlobalFiler PCR Amplification Kit on its own (blue) following the standard GlobalFiler manufacturer’s protocol.

**Figure 6 genes-15-01199-f006:**
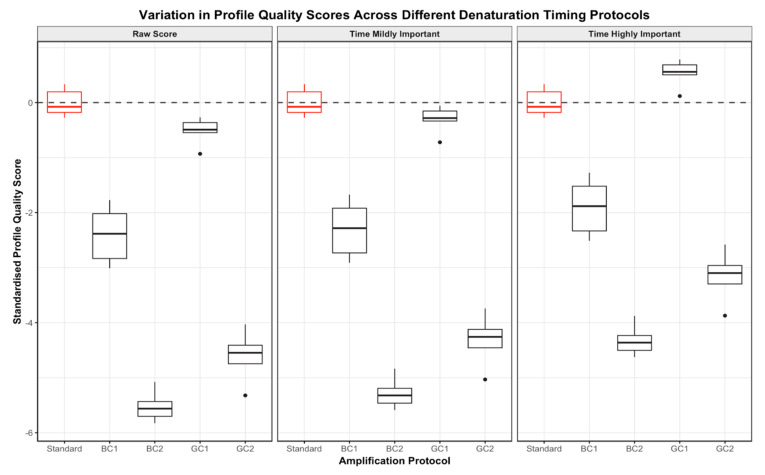
Boxplots demonstrating the spread in standardised profile quality scores across the bulk (BC) and gradient (GC) denaturation timing change protocols and standard GlobalFiler protocol for DNA profiles produced using the GlobalFiler and Investigator Quantiplex Pro Combination Kit only. For each protocol, the spread of raw profile quality scores (**left**), scores where time was considered mildly important (**centre**) and scores where time was considered highly important (**right**) are directly compared.

**Figure 7 genes-15-01199-f007:**
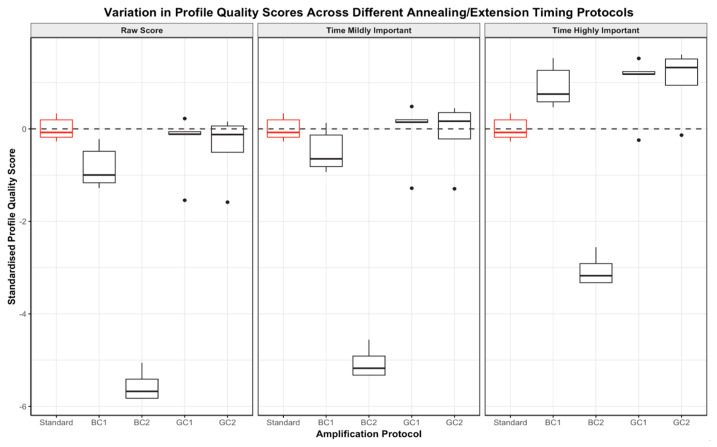
Boxplots demonstrating the spread in standardised profile quality scores across the bulk (BC) and gradient (GC) annealing/extension timing change protocols and standard GlobalFiler protocol for DNA profiles produced using the GlobalFiler and Investigator Quantiplex Pro Combination Kit only. For each protocol, the spread of raw profile quality scores (**left**), scores where time was considered mildly important (**centre**) and scores where time was considered highly important (**right**) are directly compared.

**Figure 8 genes-15-01199-f008:**
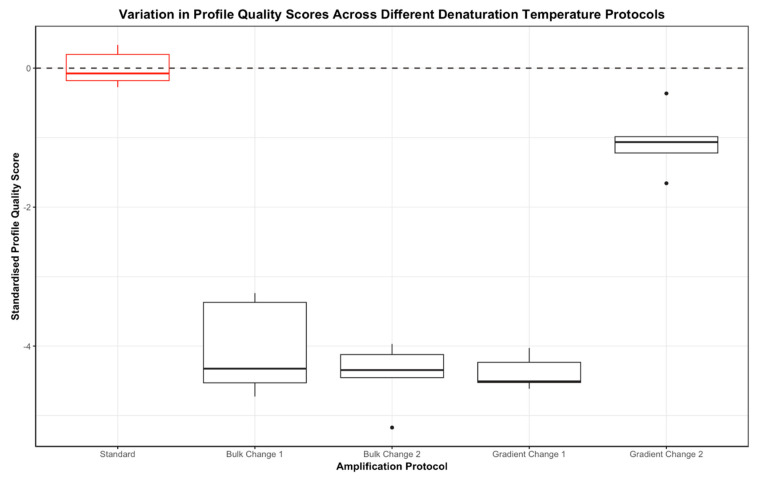
Boxplots demonstrating the spread in standardised profile quality scores across the bulk and gradient denaturation temperature change protocols and standard GlobalFiler protocol for DNA profiles produced using the GlobalFiler plus Investigator Quantiplex Pro Combination Kit only.

**Figure 9 genes-15-01199-f009:**
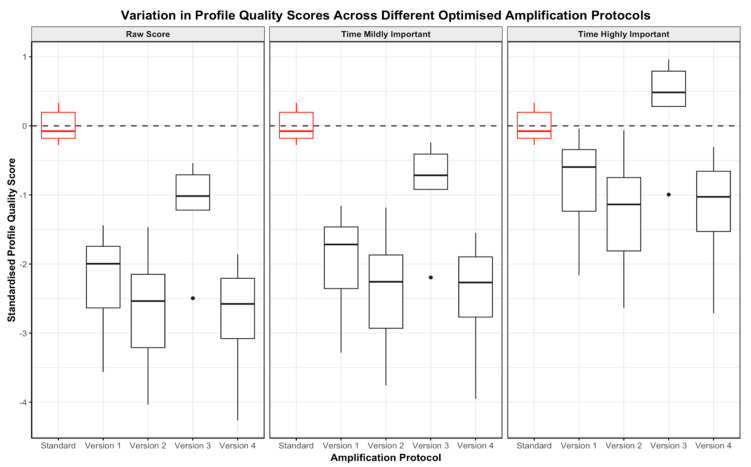
Boxplots demonstrating the spread in profile quality scores across the four optimised protocols and standard GlobalFiler protocol for DNA profiles produced using the GlobalFiler plus Investigator Quantiplex Pro Combination Kit only. For each protocol, the spread of raw profile quality scores (**left**), scores where time was considered mildly important (**centre**) and scores where time was considered highly important (**right**) are directly compared.

**Figure 10 genes-15-01199-f010:**
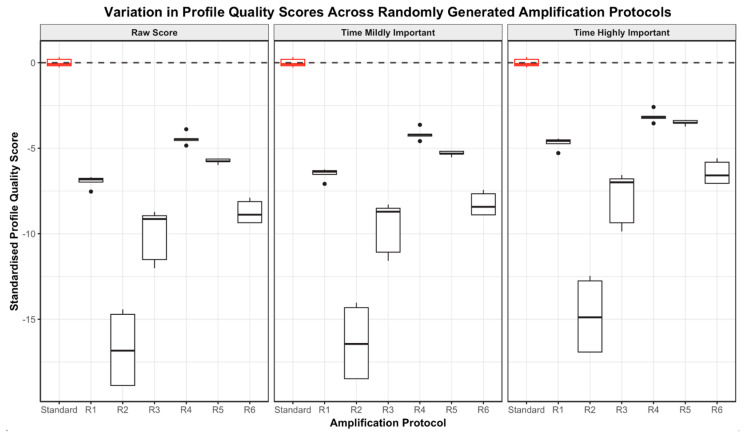
Boxplots demonstrating the spread in profile quality scores across the randomly generated protocols (R1–R6) and standard GlobalFiler protocol for DNA profiles produced using the GlobalFiler plus Investigator Quantiplex Pro Combination Kit only. For each protocol, the spread of raw profile quality scores (**left**), scores where time was considered mildly important (**centre**) and scores where time was considered highly important (**right**) are directly compared.

**Figure 11 genes-15-01199-f011:**
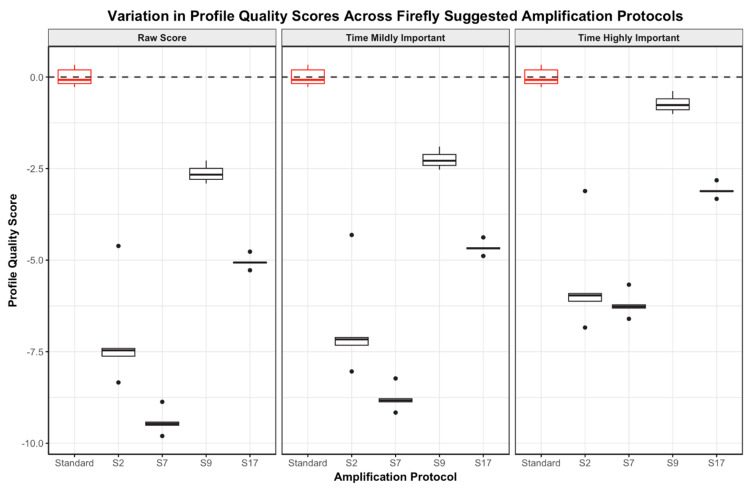
Boxplots demonstrating the spread in profile quality scores across the firefly-suggested protocols and standard GlobalFiler protocol for DNA profiles produced using the GlobalFiler plus Investigator Quantiplex Pro Combination Kit only. For each protocol, the spread of raw profile quality scores (**left**), scores where time was considered mildly important (**centre**) and scores where time was considered highly important (**right**) are directly compared.

**Table 1 genes-15-01199-t001:** PCR program changes trialled in the pilot study. Italicised conditions indicate variations from the standard GlobalFiler manufacturer’s protocol where the total number of cycles is 30. The runtimes given are calculated from the times the system holds at the required temperatures (and not from timing the physical instrument); they do not account for the time required for machine ramping and cooling between stages. Graduated changes are indicated in brackets after the factor being changed, for example, *10 s (−1 per 5 cycles)* indicates that the first 5 cycles will hold the temperature for 10 s, the following 5 cycles will hold the temperature for 9 s, etc.

Program	Cycling Conditions		Runtime
	Denaturation Step	Annealing/Extension Step	
Standard GlobalFiler manufacturer’s protocol	94 °C 10 s	59 °C 90 s	61 min
Denaturation bulk time change 1 (BC1)	94 °C 5 s	59 °C 90 s	58.5 min
Denaturation bulk time change 2 (BC2)	94 °C 2 s	59 °C 90 s	57 min
Denaturation gradient time change 1 (GC1)	94 °C *10 s (−1 per 5 cycles)*	59 °C 90 s	59.75 min
Denaturation bulk time change 1 (GC2)	94 °C *2 s (+1 per 5 cycles)*	59 °C 90 s	58.25 min
Annealing/extension bulk time change 1 (BC1)	94 °C 10 s	59 °C *60 s*	46 min
Annealing/extension bulk time change 2 (BC2)	94 °C 10 s	59 °C *30 s*	31 min
Annealing/extension gradient time change 1 (GC1)	94 °C 10 s	59 °C *90 s (−5 per 5 cycles)*	54.75 min
Annealing/extension bulk time change 1 (GC2)	94 °C 10 s	59 °C *60 s (+5 per 5 cycles)*	52.25 min
Denaturation bulk temperature change 1 (BC1)	90 °C 10 s	59 °C 90 s	61 min
Denaturation bulk temperature change 2 (BC2)	88 °C 10 s	59 °C 90 s	61 min
Denaturation gradient temperature change 1 (GC1)	*94 °C (−1 per 5 cycles)* 10 s	59 °C 90 s	61 min
Denaturation bulk temperature change 1 (GC2)	*88 °C (+1 per 5 cycles)* 10 s	59°C 90 s	61 min
Optimised version 1	94 °C *10 s (−1 per 5 cycles)*	59 °C *90 s (−5 per 5 cycles)*	53.5 min
Optimised version 2	*94 °C (−1 per 5 cycles)* 10 s	59 °C *90 s (−5 per 5 cycles)*	54.75 min
Optimised version 3	94 °C *10 s (−1 per 5 cycles)*	59 °C *60 s (+5 per 5 cycles)*	51 min
Optimised version 4	*94 °C (−1 per 5 cycles)* *10 s (−1 per 5 cycles)*	59 °C *60 s (+5 per 5 cycles)*	51 min

**Table 2 genes-15-01199-t002:** PCR cycling conditions from Table 1 expressed in parameterised terms.

Program	Denaturation Step (*D*)		Annealing/Extension Step (*AE*)
	$\mathrm{Timing}\;(t)\;\beta_{t,D,0},\;\beta_{t,D,1}$	$\mathrm{Temperature}\;(C)\;\beta_{C,D,0},\;\beta_{C,D,1}$	$\mathrm{Timing}\;(t)\;\beta_{t,AE,0},\;\beta_{t,AE,1}$
Standard GlobalFiler protocol	10, 0	94, 0	90, 0
Denaturation Timing BC1	5, 0	94, 0	90, 0
Denaturation Timing BC2	2, 0	94, 0	90, 0
Denaturation Timing GC1	10, −0.2	94, 0	90, 0
Denaturation Timing GC2	2, 0.2	94, 0	90, 0
Annealing/Extension Timing BC1	10, 0	94, 0	60, 0
Annealing/Extension Timing BC2	10, 0	94, 0	30, 0
Annealing/Extension Timing GC1	10, 0	94, 0	90, −1
Annealing/Extension Timing GC2	10, 0	94, 0	60, 1
Denaturation Temperature BC1	10, 0	90, 0	90, 0
Denaturation Temperature BC2	10, 0	88, 0	90, 0
Denaturation Temperature GC1	10, 0	94, −0.2	90, 0
Denaturation Temperature GC2	10, 0	88, 0.2	90, 0
Optimised Version 1	10, −0.2	94, 0	90, −1
Optimised Version 2	10, 0	94, −0.2	90, −1
Optimised Version 3	10, −0.2	94, 0	60, 1
Optimised Version 4	10, −0.2	94, −0.2	60, 1

**Table 3 genes-15-01199-t003:** PCR cycling conditions added to the sets of conditions shown in Table 2.

Program	Denaturation Step (*D*)		Annealing/Extension Step (*AE*)
	$\mathrm{Timing}\;(t)\;\beta_{t,D,0},\;\beta_{t,D,1}$	$\mathrm{Temperature}\;(C)\;\beta_{C,D,0},\;\beta_{C,D,1}$	$\mathrm{Timing}\;(t)\;\beta_{t,AE,0},\;\beta_{t,AE,1}$
Random 1 (R1)	4, 0.25	90, −0.10	55, −0.67
Random 2 (R2)	6, −0.15	88, 0.18	64, −0.46
Random 3 (R3)	2, 0.13	89, 0.11	63, −1
Random 4 (R4)	13, −0.20	91, −0.12	69, 0.85
Random 5 (R5)	14, −0.35	92, −0.16	43, −0.11
Random 6 (R6)	3, 0.16	88, 0.10	38, 0.42

**Table 4 genes-15-01199-t004:** PCR cycling conditions suggested by firefly algorithm. The bolded rows, marked with an asterisk, indicate the suggested PCR programs that were trialled.

Program	Denaturation Step (*D*)		Annealing/Extension Step (*AE*)
	$\mathrm{Timing}\;(t)\;\beta_{t,D,0},\;\beta_{t,D,1}$	$\mathrm{Temperature}\;(C)\;\beta_{C,D,0},\;\beta_{C,D,1}$	$\mathrm{Timing}\;(t)\;\beta_{t,AE,0},\;\beta_{t,AE,1}$
Suggested 1 (S1)	10, 0.03	94, 0.01	93, −0.02
**Suggested 2 (S2) ***	4, 0.03	94, 0.01	93, −0.02
Suggested 3 (S3)	0, 0.03	94, 0.01	93, −0.02
Suggested 4 (S4)	10, −0.22	94, 0.01	93, −0.02
Suggested 5 (S5)	0, 0.29	94, 0.01	93, −0.02
Suggested 6 (S6)	10, 0.03	94, 0.01	55, −0.02
**Suggested 7 (S7) ***	10, 0.03	94, 0.01	16, −0.02
Suggested 8 (S8)	10, 0.03	94, 0.01	93, −0.02
**Suggested 9 (S9) ***	10, 0.03	94, 0.01	55, 1.25
Suggested 10 (S10)	10, 0.03	89, 0.01	93, −0.02
Suggested 11 (S11)	10, 0.03	86, 0.01	93, −0.02
Suggested 12 (S12)	10, 0.03	94, −0.24	93, −0.02
Suggested 13 (S13)	10, 0.03	86, 0.27	93, −0.02
Suggested 14 (S14)	10, −0.22	94, 0.01	93, −1.30
Suggested 15 (S15)	10, 0.03	94, −0.24	93, −1.30
Suggested 16 (S16)	10, −0.22	94, 0.01	55, 1.25
**Suggested 17 (S17) ***	10, −0.22	94, −0.24	55, 1.25
Suggested 18 (S18)	2, 0.35	89, −0.11	48, −0.88
Suggested 19 (S19)	5, −0.16	86, 0.27	60, −0.61
Suggested 20 (S20)	0, 0.19	88, 0.15	59, −1.30
Suggested 21 (S21)	14, −2.52	90, −0.14	66, 1.06
Suggested 22 (S22)	15, −0.41	92, −0.19	33, −0.16
Suggested 23 (S23)	1, 0.24	86, 0.14	27, 0.49

## Data Availability

Additional data are available in the Appendix A document.

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
