# Peer review of "Developing a Machine Learning ‘Smart’ Polymerase Chain Reaction Thermocycler Part 2: Putting the Theoretical Framework into Practice"

_genes, 2024, doi:10.3390/genes15091199_

Round 1
Reviewer 1 Report
Comments and Suggestions for Authors
Reviewer’s comments
Developing a Machine-Learning ‘Smart’ PCR Thermocycler Part 2: Putting the Theoretical Framework into Practice
Overall, this is a well written manuscript, and the concept is novel and useful for the forensic community.
Introduction
1. Line 45 – I would put ‘(0.006 ng)’ after ‘single cell’ before ‘[4]’ reference since you are referring to the single cell.
2. Line 63 – I haven’t heard a Rapid DNA instrument called Rapid PCR. It performs all steps more quickly, not just PCR. I suggest changing all instances of ‘Rapid PCR’ to ‘Rapid DNA’.
3. Line 71 – separate ‘and or’ by a slash – ‘and/or’
4. Line 75 – add a comma after ‘i.e.’
5. Lines 87 and 88 – Change ‘Rapid PCR’ to ‘Rapid DNA’
Methods
6. Lines 144 and 145 – add a comma after ‘i.e.’
7. Line 148 – coefficient is spelled incorrectly – missing the second ‘f’
8. Line 158 – ‘CoV variation’ is redundant because the ‘V’ stands for variation. Additionally, in Part 1 of this paper, the coefficient of variation was COV, not CoV. I suggest keeping them the same. However, if you would like to change COV to CoV in the first part that’s okay too.
9. Line 177 – add an ‘s’ to ‘program’
10. Line 222 – ‘Figure 2’ should be ‘Figure 1’
11. Line 251 – ‘preliminary’ should be ‘preliminarily’
12. Line 262 – add a comma after ‘e.g.’
13. Line 279 – change ‘we’ to ‘were’
14. Line 282 – add a ‘d’ to ‘generate’
15. Lines 273-299 – add a period to the end of each process; these are complete sentences.
16. Line 301 – there is an extra space at the beginning of the sentence. Not sure if this was supposed to be indented or not.
Results
17. Lines 306-307 – what are the ‘combined PCR mixtures’? Are they the qPCR + PCR combined kit?
18. In section 3.2.1 – naming of protocols is ‘gradient change 1 protocol’, but in section 3.2.2 – name of protocols is ‘gradient change protocol 1’ – name protocols consistently throughout. Lines 411-416 in Section 3.2.2 use ‘bulk change 1 protocol’ form, so it’s inconsistent throughout.
19. Line 443 – remove ‘they’
20. Line 492 – remove ‘in’
21. Line 493 – rephrase – “…however the profiles still had sufficient information…”
22. Line 499 – add a dash between ‘real’ and ‘time’
23. Line 506 – add a comma after ‘this’
Discussion
24. Line 551 – remove ‘set’
25. Line 551 – ‘significant’ should be ‘significantly’
26. Line 552 – add a comma after ‘i.e.’
27. Lines 575-577 – this sentence doesn’t make sense, please rephrase or use punctuation.
28. Lines 596-599 – multiple thoughts in a single sentence. Please use punctuation.
29. Line 598 – add a comma after ‘it’
Tables and Figures
30. Table 1 – the information in parentheses – does the ±1 or ±5 mean seconds or cycles depending on which line it falls on? For example, ’10 seconds (-1 per 5 cycles)’ – does this mean -1 second for every 5 cycles? So cycles 1-5 would be 10 seconds, 6-10 would be 9 seconds, 11-15 would be 8 seconds, etc.?
31. Table 2 – it looks like some of the numbers in Table 1 don’t match up with some of the numbers in Table 2, unless I am misunderstanding. Please check these:
Program |
Denaturation (Timing) according to Table 1 |
Denaturation (Timing) according to Table 2 |
Annealing/Extension (Timing) according to Table 1 |
Annealing/Extension (Timing) according to Table 2 |
Denaturation Timing GC2 |
2 seconds |
10 seconds |
N/A |
N/A |
Annealing/Extension Timing GC2 |
N/A |
N/A |
60 seconds |
30 seconds |
Optimized Version 3 |
N/A |
N/A |
30 seconds |
60 seconds |
Optimized Version 4 |
N/A |
N/A |
30 seconds |
60 seconds |
32. Line 318 – Figure 3 caption – remove ‘where’
33. Lines 317 & 319 – In 317 ‘Combination kit’ is capitalized, but in 319 ‘combined kit’ is lowercase
General
34. Overuse of parentheses and ‘i.e.’
35. Make sure kit names and instrument names are consistent throughout the manuscript.
36. Why are the figure captions italicized?
37. Capitalize ‘Supplementary Material (and letter)’ throughout the text.

Comments on the Quality of English LanguageCheck thoroughly. Commas can be used to separate thoughts and make the sentence flow.
Author Response
Note from the Authors:
We are grateful to the reviewers for taking the time to read the paper and provide constructive comments, especially given its length and complexity. We address each of these comments below and note where changes have been made. These changes have also been highlighted in red in the resubmitted manuscript. We agree that these changes have improved the paper, and that it now reads more clearly than before.
Reviewer 1:
Overall, this is a well written manuscript and the concept is novel and useful for the forensic community.
Introduction
Comment: 1. Line 45 – I would put ‘(0.006 ng)’ after ‘single cell’ before ‘[4]’ reference since you are referring to the single cell.
Response: We agree this was unclear and it has now been shuffled for clarity.
Comment: 2. Line 63 – I haven’t heard a Rapid DNA instrument called Rapid PCR. It performs all steps more quickly, not just PCR. I suggest changing all instances of ‘Rapid PCR’ to ‘Rapid DNA’. .
Response: Amended.
Comment: 3. Line 71 – separate ‘and or’ by a slash – ‘and/or’ .
Response: Amended.
Comment: 4. Line 75 – add a comma after ‘i.e.’ .
Response: Amended.
Comment: 5. Lines 87 and 88 – Change ‘Rapid PCR’ to ‘Rapid DNA’ .
Response: Amended.
Methods
Comment: 6. Lines 144 and 145 – add a comma after ‘i.e.’ .
Response: Amended.
Comment: 7. Line 148 – coefficient is spelled incorrectly – missing the second ‘f’ .
Response: Amended.
Comment: 8. Line 158 – ‘CoV variation’ is redundant because the ‘V’ stands for variation. Additionally, in Part 1 of this paper, the coefficient of variation was COV, not CoV. I suggest keeping them the same. However, if you would like to change COV to CoV in the first part that’s okay too. .
Response: Thank you for pointing this out. We have amended all to COV to ensure consistency between the papers and removed the redundant “variation”.
Comment: 9. Line 177 – add an ‘s’ to ‘program’ .
Response: Amended.
Comment: 10. Line 222 – ‘Figure 2’ should be ‘Figure 1’ .
Response: Amended.
Comment: 11. Line 251 – ‘preliminary’ should be ‘preliminarily’ .
Response: Amended.
Comment: 12. Line 262 – add a comma after ‘e.g.’ .
Response: Amended.
Comment: 13. Line 279 – change ‘we’ to ‘were’ .
Response: Amended.
Comment: 14. Line 282 – add a ‘d’ to ‘generate’ .
Response: Amended.
Comment: 15. Lines 273-299 – add a period to the end of each process; these are complete sentences. .
Response: Amended at the end of each process.
Comment: 16. Line 301 – there is an extra space at the beginning of the sentence. Not sure if this was supposed to be indented or not. .
Response: No this was not intentional and has been amended.
Results
Comment: 17. Lines 306-307 – what are the ‘combined PCR mixtures’? Are they the qPCR + PCR combined kit? .
Response: Yes, we are referring to the qPCR + PCR combined kit here. We agree this was unclear and have added to line 307 to clarify.
Comment: 18. In section 3.2.1 – naming of protocols is ‘gradient change 1 protocol’, but in section 3.2.2 – name of protocols is ‘gradient change protocol 1’ – name protocols consistently throughout. Lines 411-416 in Section 3.2.2 use ‘bulk change 1 protocol’ form, so it’s inconsistent throughout. .
Response: Thank you for highlighting this for us, the manuscript has now been altered to ensure the naming of protocols is consistent throughout.
Comment: 19. Line 443 – remove ‘they’ .
Response: Removed.
Comment: 20. Line 492 – remove ‘in’ .
Response: Removed.
Comment: 21. Line 493 – rephrase – “…however the profiles still had sufficient information…” .
Response: Amended.
Comment: 22. Line 499 – add a dash between ‘real’ and ‘time’ .
Response: Amended.
Comment: 23. Line 506 – add a comma after ‘this’ .
Response: Amended.
Discussion
Comment: 24. Line 551 – remove ‘set’.
Response: Removed.
Comment: 25. Line 551 – ‘significant’ should be ‘significantly’ .
Response: Amended.
Comment: 26. Line 552 – add a comma after ‘i.e.’ .
Response: Amended.
Comment: 27. Lines 575-577 – this sentence doesn’t make sense, please rephrase or use punctuation. .
Response: Lines 568-572 have now been amended to clarify what was meant here.
Comment: 28. Lines 596-599 – multiple thoughts in a single sentence. Please use punctuation. .
Response: Appropriate punctuation has been added to this sentence to clarify between the different thoughts here.
Comment: 29. Line 598 – add a comma after ‘it’ .
Response: Amended.
Tables and Figures
Comment: 30. Table 1 – the information in parentheses – does the ±1 or ±5 mean seconds or cycles depending on which line it falls on? For example, ’10 seconds (-1 per 5 cycles)’ – does this mean -1 second for every 5 cycles? So cycles 1-5 would be 10 seconds, 6-10 would be 9 seconds, 11-15 would be 8 seconds, etc.? .
Response: – yes this is the correct interpretation of the table. We have added some material to the caption to make this clearer
Comment: 31. Table 2 – it looks like some of the numbers in Table 1 don’t match up with some of the numbers in Table 2, unless I am misunderstanding. Please check these. .
Response: We apologise for this error; these have now been corrected accordingly.
Comment: 32. Line 318 – Figure 3 caption – remove ‘where’ .
Response: Amended.
Comment: 33. Lines 317 & 319 – In 317 ‘Combination kit’ is capitalized, but in 319 ‘combined kit’ is lowercase Amended.
General
Comment: 34. Overuse of parentheses and ‘i.e.’ – .
Response: we have reduced the use of this term
Comment: 35. Make sure kit names and instrument names are consistent throughout the manuscript. – .
Response: we have gone through and tried to standardise the terms we use
Comment: 36. Why are the figure captions italicized? .
Response: We apologise for this formatting error; they have now been appropriately formatted.
Comment: 37. Capitalize ‘Supplementary Material (and letter)’ throughout the text. .
Response: This has been amended in across the manuscript.
Reviewer 2 Report
Comments and Suggestions for Authors
Dear Authors,
This is an interesting article trying to optimize the PCR conditions for application to trace DNA. I have some comments:
1. You refer to the Part 1 of this project, but what is the article associated with it? When I check the references, it could be 17 and 19.
2. The discussion only reflects on the results. Please, include comparisons with previous works from other authors/attempts (also, you can refer to the RapidPCR), and the impact of your findings in the current practice, moreover with trace DNA.
Thank you so much.
Author Response
This is an interesting article trying to optimize the PCR conditions for application to trace DNA. I have some comments:
Comment: 1. You refer to the Part 1 of this project, but what is the article associated with it? When I check the references, it could be 17 and 19.
Response: When we refer to Part 1 of this project, we are referring to another manuscript that was submitted with this paper. Part 1 outlines the theory behind the system and proof of concept studies.
Comment: 2. The discussion only reflects on the results. Please, include comparisons with previous works from other authors/attempts (also, you can refer to the RapidPCR), and the impact of your findings in the current practice, moreover with trace DNA. –
Response: we have added some material to the discussion as requested
Thank you so much.
Reviewer 3 Report
Comments and Suggestions for Authors
Review comments for “Developing a Machine-Learning ‘Smart’ PCR Thermocycler Part 2: Putting the Theoretical Framework into Practice”. This paper is very well organized, and the writing is clear and easy to follow, for the most part. It is a somewhat complex subject, even with an appropriately limited number of variables, so the reader needs to keep up with changes to conditions that are being presented. I think the authors have done a good job in helping the reader to keep the variable under discussion straight, for that I commend them. I have offered comments in areas that I think need some attention to better help the reader grasp all of the details of information being conveyed. Some are merely wording suggestions.
Introduction:
Line 35 The first use of “deoxyribonucleic acid” should be fully used prior to the acronym, as is done for PCR.
Line 57 The sentence beginning “Despite the many advancements in PCR” is a good point and one that is directly relevant to the goals of the paper. For that reason, it might be a useful addition to briefly enumerate a few of those types of advances (rxn volume, rxn mix components, pre-treatment/repair of samples, etc.) in the previous sentence, which would then directly lead in to the one above.
Line 66 I believe your references [2] and [12] are the same: McDonald, C.; Taylor, D.; Linacre, A. PCR in Forensic Science: A Critical Review. Genes (Basel) 2024, 15, 438.
Methods:
Line 134 It would be of interest to the reader to know what your injection settings were for the 3500 electrophoresis runs.
Line 218 “At the time of starting the experiment” is somewhat awkward for the reader, especially as the previous paragraphs were using “time” as a measured variable and thus, a contextually significant word. Perhaps just “At the start of the experiment”? “At the beginning of the experiment”?
Line 222 I believe you mean “Figure 1” here.
Line 246 It would be very helpful to the reader if you could give an example of how the non-zero values for the slope term for the gradient changes were determined. This would help to interpret the data shown in Table 2. The bulk change values are already intuitive and need no further explanation.
Line 265 Figure 2 is very clever, it illustrates your point well. Out of curiosity, are the data in Step 4 real, or example place holders? I admit that I did not try to match them with the information in the tables.
Results:
Lines 314 What is the point of the shaded area in Figure 3? The graph seems clear enough without it and the figure legend does not provide any details beyond the bit in the text. Is it to indicate that a small number of profiles had higher average peaks with the combined PCR mixture than just GlobalFiler alone? It is unclear what is being conveyed with the shading. Otherwise, the figure is quite clear.
Lines 326 – 330 This sentence is quite long and should be broken up to make the point clearer. In particular, the last part “prior to the start of the PCR that it will not provide useable real-time information” is difficult to understand. Perhaps “It can be seen that the inflection point of the quantification reaction has not been reached. There is a risk that even using a two-tube system (where the quantification and STR amplification reactions are in separate vessels) without 10 cycles of pre-amplification of the quantification reaction prior to the start of the accompanying/concurrent/concomitant GlobalFiler PCR will not provide useable real-time information.” or something to that effect.
Line 330 - 331 This sentence states “This fact is shown in Figure 4 where the quantification reaction is run under GlobalFiler PCR conditions” however, the legend for Figure 4 states that “amplification curve generated using the Investigator Quantiplex Pro Kit following the manufacturer’s 40-cycle protocol”. It is also specified in Line 324. These sentences seem at odds with each other. The dashed red line indicating the number of cycles in GlobalFiler is clear, but were the data shown run under those parameters?
Line 338 The sentence should probably read “combination were compared”.
Lines 349 – 352 This data may need some more explanation. It seems odd that the GlobalFiler kit alone reached maximum RFUs by cycle 9 and did not increase thereafter, while the combined kits had a far higher RFU max value, but the green line for the Quantiplex is at zero, indicating no detectable amplification for the Quantiplex alone. Why is the combo so much higher when, presumably, the Quantiplex is contributing nothing?
Lines 363 – 367 This sentence would be easier to understand if it were turned into two sentences, as there is a lot of information here. Perhaps just “When the PCR runtime was considered mildly important in quality assessment the programs with altered denaturation times using the combination kit were up to 29 minutes faster. The bulk change programs, and one gradient program (gradient change 2 protocol) still produced profiles with a significantly lower score than the standard.” This separates the points being made.
Line 372 I believe it should read “profiles produced using gradient change program 1 were now of significantly higher” as it is plural.
Lines 395 – 398 Looking at the plot for Raw Score BC1 in Figure 7 and the data in Suppl. Table 1 & 2, it is hard to see why the profiles are significantly lower quality. BC2, certainly, but BC1 seems quite close. This may bear double checking or a brief explanation. Also, p-values are given for the annealing/extension change data, but not for the denaturation change data. Is this just because the denaturation values in the box plot were so clearly separated (Line 357)?
Lines 492 – 494 This sentence is a bit awkward in its wording. Perhaps “It is important to note that all optimised PCR programs trialled produced some DNA profiles with instances of drop-out, however the profiles still contained information sufficient for downstream interpretation and analysis.”?
Lines 541 – 543 Were these expectations based on the previous experimental results, projected impact of the time differences on the scores, or all of the above?
Supplementary Tables: I noticed that some of the data values in the Suppl. Tables are in white numbers and others in black. I see no pattern in these colors, was it just for ease of viewing? I only ask as the tables have very specific color coding and I wanted to be sure that the value number colors were not meant to indicate information such as statical significance.
Discussion:
Lines 575 – 577 This sentence is a bit awkward in its wording. Perhaps “In our pilot study where we ‘unsmartly’ altered PCR program conditions, we found that using a combined kit system with the goal of PCR being conducted in a shorter time frame without compromising DNA profile quality, could be successfully approached.” “Unsmartly” = 😊.
Lines 596 – 599 Regarding the fluorescence data mentioned in this sentence, I think it would be informative to the reader to include at least one electropherogram from the GlobalFiler/Quantiplex results described in the Supplemental Tables. This would allow the reader to appreciate what the combined kit’s peak signal looked like, which is probably no real difference from a standard protocol electropherogram but solidifies your points. It should be placed in the Supplemental section as well.
Line 613 This sentence seems incomplete. Looking at the rate of change will result in what?
I noticed that amongst your parameter alterations you did not change the anneal/extension temperature. Is this because it was likely to have a more profound effect on the quality of the profiles than the other parameters, especially artefacts which were not assessed or that it was unlikely to affect the time of the PCR, which was your primary focus in the study? I think it was wise not to do so given the scope of the study, simply curious. I enjoyed reviewing this paper.

Author Response
Reviewer 3:
This paper is very well organized, and the writing is clear and easy to follow, for the most part. It is a somewhat complex subject, even with an appropriately limited number of variables, so the reader needs to keep up with changes to conditions that are being presented. I think the authors have done a good job in helping the reader to keep the variable under discussion straight, for that I commend them. I have offered comments in areas that I think need some attention to better help the reader grasp all of the details of information being conveyed. Some are merely wording suggestions.
Introduction:
Comment: Line 35 The first use of “deoxyribonucleic acid” should be fully used prior to the acronym, as is done for PCR.
Response: Thank you for pointing this out, it has now been amended accordingly.
Comment: Line 57 The sentence beginning “Despite the many advancements in PCR” is a good point and one that is directly relevant to the goals of the paper. For that reason, it might be a useful addition to briefly enumerate a few of those types of advances (rxn volume, rxn mix components, pre-treatment/repair of samples, etc.) in the previous sentence, which would then directly lead into the one above. –
Response: we have added a sentence as suggested listing some of these enhancements.
Comment: Line 66 I believe your references [2] and [12] are the same: McDonald, C.; Taylor, D.; Linacre, A. PCR in Forensic Science: A Critical Review. Genes (Basel) 2024, 15, 438.
Response: Thank you for highlighting this, we have amended and renumber the references to remove the duplicate.
Methods:
Comment: Line 134 It would be of interest to the reader to know what your injection settings were for the 3500 electrophoresis runs.I –
Response: we have added this information to the paper
Comment: Line 218 “At the time of starting the experiment” is somewhat awkward for the reader, especially as the previous paragraphs were using “time” as a measured variable and thus, a contextually significant word. Perhaps just “At the start of the experiment”? “At the beginning of the experiment”?
Response: We agree this was confusing and have amended this line for clarity.
Comment: Line 222 I believe you mean “Figure 1” here.
Response: Amended.
Comment: Line 246 It would be very helpful to the reader if you could give an example of how the non-zero values for the slope term for the gradient changes were determined. This would help to interpret the data shown in Table 2. The bulk change values are already intuitive and need no further explanation. –
Response: we have added an explanatory example to the text hear to assist with understanding
Comment: Line 265 Figure 2 is very clever, it illustrates your point well. Out of curiosity, are the data in Step 4 real, or example place holders? I admit that I did not try to match them with the information in the tables.
Response: Thank you! We appreciate that a figure like this is necessary for readers who may have little understanding of machine-learning. The data shown in Step 4 are example place holders but do closely mimic the values that were truly generated in this study.
Results:
Comment: Lines 314 What is the point of the shaded area in Figure 3? The graph seems clear enough without it. and the figure legend does not provide any details beyond the bit in the text. Is it to indicate that a small number of profiles had higher average peaks with the combined PCR mixture than just GlobalFiler alone? It is unclear what is being conveyed with the shading. Otherwise, the figure is quite clear. –
Response: we have added some information in the caption to make it clear what the shaded area refers to
Comment: Lines 326 – 330 This sentence is quite long and should be broken up to make the point clearer. In particular, the last part “prior to the start of the PCR that it will not provide useable real-time information” is difficult to understand. Perhaps “It can be seen that the inflection point of the quantification reaction has not been reached. There is a risk that even using a two-tube system (where the quantification and STR amplification reactions are in separate vessels) without 10 cycles of pre- amplification of the quantification reaction prior to the start of the accompanying/concurrent/concomitant GlobalFiler PCR will not provide useable real-time information.” or something to that effect.
Response: Thank you for the suggestion and we agree that this was unclear. The section has now been rewritten to improve clarity for the reader.
Comment: Line 330 - 331 This sentence states “This fact is shown in Figure 4 where the quantification reaction is run under GlobalFiler PCR conditions” however, the legend for Figure 4 states that “amplification curve generated using the Investigator Quantiplex Pro Kit following the manufacturer’s 40-cycle protocol”. It is also specified in Line 324. These sentences seem at odds with each other. The dashed red line indicating the number of cycles in GlobalFiler is clear, but were the data shown run under those parameters?
Response: Thank you for bringing this to our attention, this is a typo and we apologise. Figure 4 demonstrates a “standard” quantification for 0.5 ng of DNA, not a GlobalFiler PCR. The appropriate sections have been amended to correctly describe the data.
Comment: Line 338 The sentence should probably read “combination were compared”. Amended.
Comment: Lines 349 – 352 This data may need some more explanation. It seems odd that the GlobalFiler kit alone reached maximum RFUs by cycle 9 and did not increase thereafter, while the combined kits had a far higher RFU max value, but the green line for the Quantiplex is at zero, indicating no detectable amplification for the Quantiplex alone. Why is the combo so much higher when, presumably, the Quantiplex is contributing nothing? –
Response: We are not sure of why this pattern has occurred and we are now starting to move the project more info the phase of investigating the fluorescent real-time feedback. We have added some text below this image that reflects this current state of uncertainty.
Comment: Lines 363 – 367 This sentence would be easier to understand if it were turned into two sentences, as there is a lot of information here. Perhaps just “When the PCR runtime was considered mildly important in quality assessment the programs with altered denaturation times using the combination kit were up to 29 minutes faster. The bulk change programs, and one gradient program (gradient change 2 protocol) still produced profiles with a significantly lower score than the standard.” This separates the points being made.
Response: This section has been edited for clarity and we thank the reviewer for the suggestion.
Comment: Line 372 I believe it should read “profiles produced using gradient change program 1 were now of significantly higher” as it is plural.
Response: You are correct, this has been amended.
Comment: Lines 395 – 398 Looking at the plot for Raw Score BC1 in Figure 7 and the data in Suppl. Table 1 & 2, it is hard to see why the profiles are significantly lower quality. BC2, certainly, but BC1 seems quite close. This may bear double checking or a brief explanation. Also, p-values are given for the annealing/extension change data, but not for the denaturation change data. Is this just because the denaturation values in the box plot were so clearly separated (Line 357)?
Response: We agree that the difference in Raw Score between BC1 and the standard profiles is interesting. The data has been checked and this result is correct. The features that are contributing to the statistical significantly different profile qualities are the peak heights (much smaller in BC1) and the COV (much less balance and more drop out in BC1). You are correct in that the p-values were not given for the denaturation change data as they were so clearly separated in the boxplots and we wanted to avoid ‘clogging’ the discussion with a series of p-values.
Comment: Lines 492 – 494 This sentence is a bit awkward in its wording. Perhaps “It is important to note that all optimised PCR programs trialled produced some DNA profiles with instances of drop-out, however the profiles still contained information sufficient for downstream interpretation and analysis.”?
Response: We agree this improves the clarity of the manuscript and have amended.
Comment: Lines 541 – 543 Were these expectations based on the previous experimental results, projected impact of the time differences on the scores, or all of the above?
Response: These expectations were based on all of the above, with theoretical understanding of the reaction, previous results and projected impacts all playing a role in this decision. This section has been amended to clarify this.
Comment: Supplementary Tables: I noticed that some of the data values in the Suppl. Tables are in white numbers and others in black. I see no pattern in these colors, was it just for ease of viewing? I only ask as the tables have very specific color coding and I wanted to be sure that the value number colors were not meant to indicate information such as statical significance.
Response: The colours of the boxes and numbers in the Supplementary Tables was indicative of the scores, relative to the other scores within each category (Raw, Time Mildly Important and Time Highly Important). They have no relation to statistical significance.
Discussion:
Comment: Lines 575 – 577 This sentence is a bit awkward in its wording. Perhaps “In our pilot study where we ‘unsmartly’ altered PCR program conditions, we found that using a combined kit system with the goal of PCR being conducted in a shorter time frame without compromising DNA profile quality, could be successfully approached.” “Unsmartly” = ? .
Response: we have reworded this sentence and broken it up to make it clearer.
Comment: Lines 596 – 599 Regarding the fluorescence data mentioned in this sentence, I think it would be informative to the reader to include at least one electropherogram from the GlobalFiler/Quantiplex results described in the Supplemental Tables. This would allow the reader to appreciate what the combined kit’s peak signal looked like, which is probably no real difference from a standard protocol electropherogram but solidifies your points. It should be placed in the Supplemental section as well. –
Response: we have added an example of an combination system amplified profile into the supplementary material for the paper.
Comment: Line 613 This sentence seems incomplete. Looking at the rate of change will result in what? –
Response: we have changed this sentence
I noticed that amongst your parameter alterations you did not change the anneal/extension temperature. Is this because it was likely to have a more profound effect on the quality of the profiles than the other parameters, especially artefacts which were not assessed or that it was unlikely to affect the time of the PCR, which was your primary focus in the study? I think it was wise not to do so given the scope of the study, simply curious
Response: Yes, we chose to not change the anneal/extension temperature for two reasons. Firstly, we knew the polymerase enzymes used in the PCR kits have a very narrow range of optimal temperatures, so we did not want to make a change that we knew would profoundly alter the quality of the profiles. Secondly, you are correct that it also did not align with the specific goal of this study (reducing the PCR time).
Round 2
Reviewer 3 Report
Comments and Suggestions for Authors
All of my comments were satisfactorily addressed in the revised manuscript. I noticed that in Lines 397 - 407 there are some references to supplementary data tables where the word "table" was not capitalized. You may wish to check on those if that is required. Otherwise I think the paper is complete.